# The Story of Sadāprarudita's Search for Dharma and the Worship of the *Prajñāpāramitā Sūtra* from India to Sixth-Century China

**Wen Zhao**

College of Philosophy, Nankai University, Tianjin 300350, China; wenzhao358@163.com

**Abstract:** The story of bodhisattva Sadāprarudita's search for Dharma in the *Prajñāpāramitā Sūtra* has served to successfully shape the characters of the Dharma seeker, bodhisattva Sadāprarudita, and the Dharma preacher (*dharmabhāṇakas*), bodhisattva Dharmodgata. This narrative carried much information about the veneration of the *Prajñāpāramitā Sūtra* in Indic contexts, and it also enthused Chinese Buddhists of the sixth century CE to create the *Prajñāpāramitā Sūtra* written in gold calligraphy. Emperor Wu of the Liang organized *pañcavārṣika* assemblies centred on the lectures and veneration of the gold-calligraphy Sūtra, and the Tiantai master Huisi made a vow to create such a scroll around the same time. In the relevant accounts, Chinese preachers are always associated with the Dharma preacher Dharmodgata in the narrative, which in turn enhanced their authority in the contexts in which they operated. The narrative thus helped to promote the transmission of the text across the cultural boundaries in which the Dharma preacher, as the embodied agent of the *Prajñāpāramitā* text, played a significant role.

**Keywords:** *Prajñāpāramitā Sūtra*; Dharma preacher; Emperor Wu of Liang; Master Huisi

## 1. The Worship of the *Prajñāpāramitā Sūtra* in Early Mahāyāna Buddhism and the Narrative of Seeking Prajñāpāramitā

The frequent occurrence of relics and *stūpa*s, the monuments in which relics are kept and honoured, in early Mahāyāna texts, and particularly in the *Prajñāpāramitā Sūtra*, has garnered much attention in scholarship. Hirakawa (1963) sees *stūpa* sites as the primary institutional base of the early Mahāyāna. However, based on textual evidence found in Mahāyāna literature, Schopen (1975, pp. 170, 179) argues that early Mahāyānists rejected the veneration of the *stūpa* and relics and instead developed new places of worship named *caityabhūta*, where Mahāyāna texts were to be memorized, recited, written and taught. The term *caitya* (shrine) can refer to a *stūpa*, to a *bodhimaṇḍa* (the place where Buddhas sit on the night they attain Buddhahood) or to other places associated with the life of the Buddha. Nevertheless, Drewes (2007) contradicts Schopen on this point,[1] arguing that the comparison between the *stūpa* and the places where *sūtra*s were recited is nothing but a simile, a rhetorical strategy that can be found in both Mahāyāna and non-Mahāyāna literature. For instance, one interesting case from a Mahāyāna text indicates that the Buddha-to-be in his mother's womb is comparable to the relics in a *stūpa* (Drewes 2007, pp. 107–8). In Mahāyāna literature, the merit of reciting, copying and preaching Mahāyāna texts is emphasized by comparing them with the relatively smaller amount of merit generated by paying respect and giving donations to the *stūpa* and relics.

Merit (*punya*) is a fundamental aspect of Buddhist ethics across all traditions. Discussions concerning the creation of merit are hence abundant in Buddhist literature, and one of the most significant merit-making deeds is paying respect to the three jewels: the Buddha, Dharma and Saṅgha (Buddhist order or community). Nevertheless, in Mahāyāna literature, as Tanabe (2004) notes, the notion of merit is extended to the idea of benefits obtained by means of some ritual actions related to Mahāyāna Sūtras.[2] If we read through the

*Aṣṭasāhasrikā Prajñāpāramitā* (Perfection of Wisdom in 8000 lines, which contains chapters I to XXXII) as a whole, such merits of taking up, reciting and writing the *Prajñāpāramitā* text are repeated in chapters III to XII more frequently than in the other chapters. In chapters III to XII, many discussions are devoted to comparing these ritual actions with other well-known beneficial religious practices, arguing, for instance, that their merit exceeds that of offering to *stūpa*s (III) or relics (IV) whilst affirming the *Prajñāpāramitā* as the great incantation (*vidyā*) and declaring its worldly benefits (III) as being greater than other Buddhist teachings (V), etc. Likewise, one who criticizes the *Prajñāpāramitā* is said to go to hell (VII), and one who does not recite the *Prajñāpāramitā* text correctly is deemed to be under the influence of Māra (XI), etc. As will be seen in the discussions below, such statements were not merely regarded as a form of literary expression but as clear exhortations to the Buddhist community to venerate the *Prajñāpāramitā Sūtra*.

The title "8000 lines" is actually a later classification. Karashima (2011, p. 1, n. 1) assumes that the earliest Chinese translation by Lokakṣema 支婁迦讖, dating to 179–180 CE, was originally entitled *Banruoboluomi jing* 般若波羅蜜經 (Skt. *Prajñāpāramitā*) or *Mohe banruoboluomi jing* 摩訶般若波羅蜜經 (Skt. *Mahāprajñāpāramitā*), and that by adding the name of its first chapter "Daoxing pin" 道行品, the title was thereafter changed to *Daoxing banruo jing* 道行般若經. The translation of this text was probably based on an original text in Gāndhārī (Karashima 2013). In and around the first half of the 3rd century, the original *Prajñāpāramitā* was further expanded into the "*Larger Prajñāpāramitā*", since sometime after 260 CE, Zhu Shixing 朱士行 heard of the existence of the *Larger Prajñāpāramitā* and set off on a long journey to Khotan.[3] The "*Larger Prajñāpāramitā*" refers to a group of Prajñāpāramitā texts of different sizes: the *Śatasāhasrikā* (100,000 lines), *Pañcaviṃśatisāhasrikā* (25,000 lines) and *Aṣṭadaśasāhasrikā* (18,000 lines). This classification according to the size of the texts denotes the number of metrical units, with one metrical unit "line" (the metrical unit applied to prose) containing 32 syllables (*śloka*) (Zacchetti 2015, p. 176). In outlining the process by which the *8000 lines* was expanded to form the *Larger Prajñāpāramitā*, Zacchetti writes "The adoption of lists [of terms] as a key expository strategy is a prominent feature of the *Larger Prajñāpāramitā.* This form of exposition, which is found in the *Aṣṭasāhasrikā* in an embryonic form, is used systematically in *Larger Prajñāpāramitā* texts." (Zacchetti 2015, p. 184). Following the *8000 lines* and *Larger Prajñāpāramitā*, a series of shorter *Prajñāpāramitā* texts, consisting of condensed summaries of these larger texts, also appeared, including the *Vajracchedikā Prajñāpāramitā* (Diamond-Cutter Perfection of Wisdom), *Suvikrāntavikrāmiparipṛcchā Prajñāpāramitā* (Perfection of Wisdom Requested by Suvikrāntavikrāmin) and the *Saptaśatikā Prajñāpāramitā* (Perfection of Wisdom in 700 lines), etc.[4]

In this section, I would like to first examine the internal and external evidence concerning the earliest development of the veneration of the *Prajñāpāramitā* in India and the role that the Dharma preacher (*dharmabhāṇaka*) played in this process. In particular, one narrative of bodhisattva Sadāprarudita's search for Dharma, which is preserved in some versions of the text in *8000 lines* and the *Larger Prajñāpāramitā* (Zhao 2020, p. 254), may have contributed significantly to promoting the veneration of the *Prajñāpāramitā* text and the authority of its Dharma preacher.

Internal evidence concerning the veneration of the *Prajñāpāramitā* can be traced back to the earliest versions of the *Prajñāpāramitā Sūtra*. As Karashima (2013, p. 181) has pointed out, in the narrative of bodhisattva Sadāprarudita in the *Perfection of Wisdom in 8000 lines* (including its earliest Chinese translation, the *Daoxing banruo jing*), the destination of Dharma seeking, the country/city of Gandhāvatī, seems to hint at Gandhāra. In 1999, in the Bajaur region of north-western Pakistan, birch bark manuscript fragments, written in the Gāndhārī language and the Kharoṣṭhī script, of the *Prajñāpāramitā Sūtra* were unearthed. The date and contents of these fragments are quite close to those of the *Daoxing banruo jing* 道行般若經 by Lokakṣema (Falk and Karashima 2012), and the latter is very likely translated from a Gāndhārī original (Karashima 2013). Thus, along with the dissemination of

the Gāndhārī *Prajñāpāramitā* text, presumably during the Kuṣāṇa period, it would appear that the worship of the *Prajñāpāramitā Sūtra* was already present in Gandhāra.

According to the narrative of bodhisattva Sadāprarudita from the *Daoxing banruo jing*, a copy of the *Prajñāpāramitā Sūtra*, written on a gold tablet, is kept in a container made of seven gems in a pavilion (Ch. *tai* 臺) of the city Gandhāvatī:

> There is a bodhisattva named Dharmodgata, the most honorable one among human beings. Everybody serves him and pays homage to him. For the sake of the *Prajñāpāramitā Sūtra*, the bodhisattva built such a pavilion. Therein, there is a *Sūtra*-container made of the seven gems. Using the best gold as the writing material (*su* 素 literally means "white silk"), he wrote the *Prajñāpāramitā Sūtra* on it. In the container, there are many hundreds of kinds of rare incense. The bodhisattva Dharmodgata makes offerings to [the *Sūtra*] everyday, employing diverse flowers and famous incense, lighting lamps and hanging banners. A baldachin with a variety of precious substances and many hundred kinds of music are dedicated to the *Prajñāpāramitā Sūtra*.[5]

The prototype of the *Sūtra* in this passage could be connected with the archaeological finding of a fragmentary 9th-century text of the *Larger Prajñāpāramitā* (paralleling the first *Abhisamaya of 25,000 lines*) from Sri Lanka, which is inscribed on seven gold leaves of considerable size (von Hinüber 1983). In addition, a number of small fragments of *Perfection of Wisdom in 25,000 lines* inscribed on copper plaques were also discovered in the remains of a *stūpa* in Sri Lanka in 1923 (Paranavitana 1933, p. 200; Zacchetti 2015, p. 188), which reflect the influence of the conception of the Dharma relic (see below). However, we can assume that the gold or copper plates with Kharoṣṭhī inscriptions made in the first century CE[6] share more physical features with the original type of the gold tablet *Prajñāpāramitā Sūtra* mentioned in this passage, since the earliest *Prajñāpāramitā Sūtra* might have been composed in Gāndhārī around the same period in Gandhāra (Karashima 2013).

The circumstances of the *Sūtra's* veneration are described in more detail in the Sanskrit version of *8000 lines*, presenting a modified ritual that differs from the one which was enacted in the Gandhāran area:

> The bodhisattva Dharmodgata had at that time built, for the perfection of wisdom, a peaked house[7] which was made of seven precious substances, decorated with saffron, and surrounded by ornaments of pearls. On the four corners of the peaked house, *Maṇi* jewels were placed, which served as the lamps. The incense pots were suspended at the four directions, in which pure black agarwoods were perfuming. And in the middle of that peaked house a couch made of the seven precious substances was put up, and thereupon a container made of four large gems. Into that the perfection of wisdom was placed, written with melted *vaiḍūrya* on golden tablets. The peaked houses were decorated with various hanging strips and garlands.[8]

Compared with the *Daoxing banruo jing*, this passage supplies additional information concerning the arrangement of the sacred space where the Sūtra's veneration takes place. Although this record of the sacred space is found in the narrative, it may not merely be literary description; it could also reflect the dimensions of the *Prajñāpāramitā Sūtra's* veneration before the 5th century in the Gupta period (319–550 CE), since Kumārjīva's translation (T. 227), completed at the beginning of 5th century, is a very close parallel to this Sanskrit recension.[9]

The earliest external evidence (i.e., archaeological or textual sources beyond *Prajñāpāramitā* literature) concerning the veneration of *Prajñāpāramitā* texts in India can be found in the travel report of Faxian 法顯 (ca. 337–422 CE), composed in the early 5th century. When Faxian visited Mathurā, he made the following records of the *stūpa*s among the local Buddhist community:

> The *stūpa*s of Śāriputra, Maudgalyāyana and Ānanda, and the *stūpa*s of Abhidharma, Vinaya and Sūtra are established in the dwelling place of the Buddhist

assembly. One month after *varṣa* (the rains retreat) ... after Dharma-preaching, people make offerings to the *stūpa* of Śāriputra with various flowers, lighting the lamps for all nights. The drama players are asked to display the story that Śāriputra, as a great Brāhmaṇa, visited Buddha to be converted to Buddhism. [The veneration of] Mahāmaudgalyāyana and Mahākāśyapa are the same. Buddhist nuns (*bhikṣuṇī*) always make offerings to Ānanda, since Ānanda requested the Lord to allow the ordination of women. Novice monks (*śrāmaṇera*) always make offerings to Rāhula, masters of the Abhidharma give donations to the Abhidharma, and masters of the Vinaya make offerings to the Vinaya on a specific day of each year. Mahāyāna followers make offerings to *Prajñāpāramitā*, Mañjuśrī and Avalokiteśvara, etc.[10]

Faxian visited India during the Gupta period, at a time when Mathurā had long been established as a significant Buddhist centre, already from the Kuṣāṇa period. In Faxian's accounts, the veneration of the Buddhist texts and relics (the physical remains of the Buddha or eminent monks) are deeply integrated, and the textual objects of veneration are not limited to the text of the *Prajñāpāramitā Sūtra* but also refer to "mainstream" Buddhist texts, viz., the *Abhidharma*, *Vinaya* and *Sūtras*. In addition, the *stūpa*s containing Buddhist texts occur side by side with the *stūpa*s of holy disciples such as Śāriputra, Maudgalyāyana, Mahākāśyapa and Ānanda, etc., which indicates that both "mainstream" and Mahāyāna texts were worshipped in the same way as relics. Indeed, the above case concerning the *Perfection of Wisdom in 25,000 lines* on copper plaques that was discovered in the remains of a *stūpa* in Sri Lanka testifies to the affinities between the ritual veneration of relics and *sūtras* in the Buddhist community. It moreover proves that the association between the veneration of Buddhist texts and the idea of the Dharma relic was prevalent among both "mainstream" and Mahāyāna Buddhist traditions.

With regard to the followers of Mahāyāna Buddhism, according to Conze (1978, p. 14), the idea of the "mother of Tathāgatas" in the *Prajñāpāramitā Sūtra* influenced the personification of Prajñāpāramitā as a female bodhisattva. He argues that Faxian's records prove that statues of Prajñāpāramitā can be dated in India to as early as 400 CE. However, the "Prajñāpāramitā" mentioned in Faxian's passage likely refers to *Prajñāpāramitā* scripture, just as the he refers to "mainstream" scriptures, rather than the image of a female bodhisattva. The earliest description of Prajñāpāramitā as a female bodhisattva is found in a 7th-century Chinese translation of the ritual text, the *Tuoluoni ji jing* (Ch. 陀羅尼集經, Skt. *Dhāraṇīsamuccaya Sūtra*; T. 901), which remarks on the figure of Prajñāpāramitā as well as on the relevant mantras and rituals.[11] Currently we only have surviving examples of Prajñāpāramitā's presence in India as a female bodhisattva from the 7th century CE, the earliest surviving example being an early-7th-century bronze from Gilgit, Kashmir (Chemburkar 2022).

The veneration of Buddhist texts is always accompanied with the preaching of the text. In his article "*Dharmabhāṇaka* in early Mahāyāna", Drewes (2011) opens up a fresh perspective on the composition of Mahāyāna texts by investigating a certain figure therein who is given particular place of prominence, the Dharma preacher (*dharmabhāṇaka*). Following Drewes, in a more recent study, Apple argues as follows:

Indian Buddhist cultural understandings of textual discourses resulted in individual and group domestic worship of texts, the veneration of copies of sūtras owned by *dharmabhāṇaka*s, and the veneration of *dharmabhāṇaka*s as Buddhas who embodied the dharma texts that they recited ... I will suggest for constructive consideration that the "cult of the book" was a cult of a certain type of textual culture that was both oral and written, and that, rather than being a stable or local cult phenomena, it was comprised of highly mobile and translocal textual communities who carried their object of veneration with them and kept such objects in domestic locations. (Apple 2014, p. 26)

The cult of texts and the function of the Dharma preacher in transmitting them shed new light on our understanding of passages which refer to the merits of venerating Mahāyāna Sūtras in the early works of that tradition. As Apple demonstrates, the practitioners' intention was to concurrently establish the "worship of texts" and the authority of the Dharma preacher as the textual agent.

Evidence concerning the authority of the Dharma preacher in relation to the *Prajñāpāramitā Sūtra* can also be found in the narrative of bodhisattva Sadāprarudita's search for Dharma. Karashima (2013, p. 183) has briefly mentioned that the composer of the narrative might have been a *dharmabhāṇaka(s)* in Gāndhara. In the narrative, Sadāprarudita, as a bodhisattva who has just begun his career, acquires instruction from a voice in the sky and a manifestation of the Buddha, which indicate to him the direction in which he may find the text. Subsequently he encounters the present Buddhas engaged in meditative concentration (*samādhi*). The story climaxes in a meeting with a Dharma preacher (who can be understood as a Buddha), namely, the bodhisattva Dharmodgata residing in the city Gandhāvatī. As a matter of fact, the whole course of Sadāprarudita's search for Dharma resembles the course of Buddha visualization as reflected in the *Pratyutpanna-buddhasaṃmukhāvasthitasamādhi Sūtra* 般舟三昧經, and that the story seeks to emphasize the significance of the Dharma preacher (Zhao 2020). The story was thus included in the *Prajñāpāramitā Sūtra* to establish the authority of the Dharma preacher and thereby justify its transmission.

Although we lack sufficient proof, beyond textual sources, that would substantiate the role of this Dharma preacher in the early Mahāyāna community in Indic contexts, such evidence does arise in the context of 6th-century China. At this time, Emperor Wu of Liang 梁武帝 (Xiao Yan 蕭衍, 464–549 CE), a famous Chinese Buddhist monarch, organized ceremonial Dharma assemblies in his Buddhist Palace Chapel, which included lectures centred on the *Larger Prajñāpāramitā*, emulating, in certain regards, the actions of the Dharma preacher in the narrative of Sadāprarudita. In his lectures, the emperor always used a *Prajñāpāramitā Sūtra* written in gold calligraphy, and his veneration of the text in the capital of the Liang Dynasty promoted its worship more widely in northern China. Around the same time, Master Huisi, who preached the *Prajñāpāramitā Sūtra* in the area close to the border between northern and southern China, also made a vow to create a copy of the *Sūtra* in gold calligraphy. However, as a Dharma preacher, he had chosen to treat the *Prajñāpāramitā Sūtra* in a different manner, regarding it as an incantation to resolve urgent secular issues, to which end a gold copy would stand the test of time until the arrival of the future Buddha Maitreya.

## 2. The Narrative of Sadāprarudita and Emperor Wu's Gold-Calligraphy *Prajñāpāramitā*

Translations of Mahāyāna Sūtras into Chinese were always accompanied by explanations and preaching. Lokakṣema's *Daoxing banruo jing* represents the first attempt at translating the Mahāyāna Sūtra into Chinese. Another Chinese recension of the *Aṣṭasāhasrikā Prajñāpāramitā*, the *Da mingdu jing* 大明度經 (T. 225), attributed to Zhi Qian 支謙, contains such an exposition in the form of an interlinear commentary in its first chapter. Nattier points out that the first chapter of T. 225 cannot be attributed to Zhi Qian,[12] arguing that its interlinear commentary is likely the product of the Buddhist community headed by Kang Senghui 康僧會 (?–280 CE) that was active in 3rd-century southern China (Nattier 2008, pp. 136–37). Such a commentary represents the earliest evidence for the preaching of the *Prajñāpāramitā Sūtra* in China.

Around the same period, the followers of the *Daoxing banruo jing* in the northern Chinese Buddhist centre of Luoyang 洛陽 heard about the existence of a "more complete" version of the *Prajñāpāramitā*, i.e., the *Larger Prajñāpāramitā*.[13] Thus, Zhu Shixing 朱士行 travelled from Luoyang to Khotan in search of this more complete version. The well-known legend of his experience in Khotan adopts the motif of the indestructible scripture, unburnt even by fire, which can be found in a series of early Buddhist stories (Zürcher 2007, p. 63). The Sanskrit scripture sent by him back to China was translated with the title *Fangguang jing* 放光經 in 291 CE, and slightly earlier, another scripture, the *Guangzan jing* 光讚經,

was also carried by a Khotanese monk to China and translated by Dharmarakṣa 竺法護 in 286 CE.

Due to its mutual promotion in legend and philosophy, the *Larger Prajñāpāramitā* became favoured among Buddhist monks and literati from the beginning of 4th century.[14] However, the discussions and debates surrounding the teaching of the above two recensions of the *Larger Prajñāpāramitā* came to their end after the arrival of the Kuchean monk Kumārajīva 鳩摩羅什 (343–413 CE) in Chang'an 長安 in 401 CE, who translated the *8000 lines*, the *Larger Prajñāpāramitā* and many other Buddhist texts. The *Larger Prajñāpāramitā* was re-translated in close conjunction with its commentary, the *Da zhidu lun* 大智度論 (T. 1509), which can be viewed as a result of teaching and discussions surrounding the meaning of the text during the translation process. However, after Kumārjīva's death in 413 CE, the *Prajñāpāramitā Sūtra* generally lost its attraction among the Chinese Buddhist elite, many of whom, and even the disciples of Kumārjīva themselves, turned to the theory of Buddha nature 佛性, as elaborated in the *Da banniepan jing* (Ch. 大般涅槃經, Skt. *Mahāparinirvāna Sūtra*) and to one Abhidharma treatise, the *Chengshi lun* (Ch. 成實論, Skt. *Satyasiddhi-śāstra*), which was also translated by Kumārjīva. This remained the case until Emperor Wu became dedicated to the preaching of the *Larger Prajñāpāramitā* translated by Kumārjīva.

Before Emperor Wu of Liang organized lectures on the *Prajñāpāramitā* in the *pañcavārṣika* assemblies, he had already laid significant groundwork, completing the exegesis of Kumārajīva's translation of the *Larger Prajñāpāramitā* in 512 CE. When he read through the *Sūtra*, he noticed the aforementioned narrative of Sadāprarudita's search for Dharma and highly praised the deeds of that bodhisattva, writing the following words in his "Zhujie dapin xu" 註解大品序 ("Preface to the Exegesis on the *Larger Prajñāpāramitā Sūtra*") collected in *Chu sanzang ji ji* 出三藏記集 (T. 2145):

> In ancient times, when the real teaching [of Buddha] had not been spread, in the place where the name of the Dharma had not been heard, [Sadāprarudita] continued to strive, travelling through grass and swamp and experiencing the dangers of long-distance travel. He concentrated his mind upon listening, contemplating the quietness and expecting a miracle, he perceived a voice in the sky. He sacrificed his life out of treasuring but a half stanza and sold parts of his own body out of honouring but a single sentence. He was willing to shed blood without doubt and delighted in freely donating his bone marrow [for the Dharma]—not to mention for the divine pearl of the dragon palace, the precious pavilion and the gold tablet.[15]

This passage is an abstract of the episodes from the narrative of Sadāprarudita, in which he hears a voice instructing him to seek the *Prajñāpāramitā* from the sky but is overcome with doubt. He then has a vision of a magically created Buddha who tells him of the city of Gandhavatī and the Bodhisattva Dharmodgata. Thereafter he enters into *samādhi* and sees the Buddhas of the ten directions. Embarking on his quest, he overcomes several obstacles and eventually reaches the city of Gandhavatī. There, he wishes to find a buyer to collect offerings for Dharmodgata, but his effort is hindered by the magic of Māra. In order to test Bodhisattva Sadāprarudita, Indra manifests as a Brahmin and asks Sadāprarudita to cut off his own flesh, blood and bone marrow in exchange for a dedication to the Dharma preacher, bodhisattva Dharmodgata. Then comes the bloody scene of self-immolation, which corresponds to the expression "willing to shed blood without doubt and delighting in donating the marrow without stinginess" in the quotation above: Sadāprarudita takes a knife and stabs himself in both arms so that his blood flows out. He also cuts off the flesh from his thighs and breaks his bone to extract the marrow. In addition, "the precious pavilion and gold tablet" mentioned by Emperor Wu apparently relates to one passage concerning the practice of veneration from Kumārajīva's translation of the *Prajñāpāramitā Sūtra* (a parallel to the Sanskrit passage discussed in our last section): "In the middle of the precious pavilion, there is a big couch with seven jewels. There is a container made of four gems on the couch. The real gold tablet with the *Prajñāpāramitā Sūtra* written on it is placed inside the container."[16]

Another source also testifies that Emperor Wu of Liang attached great importance to the episode of self-immolation in the narrative of Sadāprarudita. In 516 CE, Emperor Wu ordered Baochang 寶唱, a monk in charge of the Buddhist library of Baoyun Hall 寶雲殿 in Emperor Wu's Hualin Park 華林園, to edit a collection of Buddhist miracle stories titled *Jinglü yixiang* 經律異相 (T. 2121). The stories in this collection mainly derive from *Sūtras* and *Vinaya* texts, and some of them are also collected from such treatises as the *Da zhidu lun*, as well as from the Buddhist texts composed in China. The miracle stories fall into 39 categories according to the identity of the protagonist. The very first story in the third category "Bodhisattva" is titled "Satuobolun weiwenfa maixinxuesui" 薩陀波崙為聞法賣心血髓 ("Sadāprarudita sells his heart, blood and marrow to listen to the Dharma"), namely the episode of self-immolation in the narrative of Sadāprarudita.

It therefore seems quite clear that several aspects of the narrative of Sadāprarudita's search for the Prajñāpāramitā made a deep impression on Emperor Wu. He regarded bodhisattva Sadāprarudita as a model Dharma seeker, sacrificing himself to make donations to the Dharma preacher. Thus, his lecture on the *Prajñāpāramitā Sūtra* could be further interpreted as being modelled after the Dharma preacher Dharmodgata.

Chen Jinhua has focused on the *pañcavārṣika* assemblies 無遮大會 held in the Buddhist Palace Chapel, the Chongyun Hall 重雲殿, of Emperor Wu. He noticed that the Chongyun Hall served "as a lecture-hall for the emperor, who was an avid preacher on Buddhism, especially on the *Prajñāpāramitā Sūtra*" (Chen 2006, p. 53). The *Prajñāpāramitā Sūtra* could be considered as a Dharma relic, whose veneration constituted a central part of Emperor Wu's Dharma assemblies. The chanting of the *Prajñāpāramitā Sūtra*, and accompanying rituals, served also to foster the kind of psychological environment which helped bring about fervent emotion in the audience, compelling them to make lavish gifts and even enact self-immolation as a donative act (Chen 2006, pp. 45–46).

Previous studies on the Dharma assemblies of Emperor Wu of Liang have paid much attention to their social functions, such as enhancing a sense of solidarity among the populace, generating funding for Emperor Wu's charitable programmes, etc., (Chen 2006, pp. 76–77) or politically establishing a "Buddhist country" headed by the sovereign (Yan 1998). However, Murata (2020) argues that the veneration of the *Prajñāpāramitā Sūtra* in Emperor Wu's Dharma assemblies could be regarded as a reconstruction of the scene in the narrative of Sadāprarudita (see below). In this line, I would like to further point out that the records of these assemblies would appear to suggest that the very circumstances of Emperor Wu's lectures, and even the participants' reactions within the Dharma assembly, could be also connected to specific episodes or scenes from the narrative. Emperor Wu's organization of the lectures could therefore reflect his intention to identify himself as a sacred Dharma preacher of the *Prajñāpāramitā Sūtra*, as inspired by the narrative of Sadāprarudita.

First, it should be noted that the focus of the *pañcavārṣika* assemblies is not only on Emperor Wu but also on the gold copy of the *Prajñāpāramitā Sūtra*. It is also highly likely that Emperor Wu did not preach the whole text in the Dharma assemblies but one particular chapter named "Sanhui pin" 三慧品. This title, or its equivalent *Sanhui jing* 三慧經, occurs repeatedly in historical accounts of the significant Dharma assemblies, which were made to coincide with other important occasions, such as a change of regnal title (the Chinese era name for official year numbering) or Emperor Wu's ordination as a monk, etc.[17]

The *Yujiang jinzi mohe banruoboluomi jing xu* 御講金字摩訶般若波羅蜜經序 ("Preface to the *Mahāprajñāpāramitā Sūtra* written in Gold Calligraphy for use in Royal Lectures"), collected in the *Guang hongming ji* 廣弘明集 *juan* 19 (T. 2103: 52.236b21–238a7), was composed by a Southern Liang historian, Xiao Zixian 蕭子顯 (489–537 CE), after Emperor Wu's lecture in 533 CE. Xiao Zixian was the composer of the *Nan Qi shu* 南齊書 and belonged to the royal family of the Southern Qi 南齊 (479–502 CE). This preface is a statement of fact and hence to be treated as a veracious historical account. Of particular relevance here is that it clearly relates how the gold-calligraphy *Prajñāpāramitā Sūtra* and the *Sūtra's* container were produced:

> The gold calligraphy *Mahāprajñāpāramitā Sūtra* is the most honourable in the corpus of the Dharma … the emperor prefers the Mahāyāna teaching, roaming through the ocean that is the corpus of Dharma. His intention agrees with the truth/the Dao, bearing it in his mind and holding it. The majesty preaches by himself, and his words are excellent and remarkable … Unfolding the emerald fine silk, one writes the Chinese sigillary characters with gold ink on it. The equipment is decorated with priceless jewels. The brilliance of the collocation of gold and green colours even overshadows the precious Ganoderma lucidum.[18]

Accompanying the lecture on the *Prajñāpāramitā Sūtra* by Emperor Wu of Liang, there was also a veneration of the text in the *pañcavārṣika* assembly. According to Xiao Zixian's preface, the prince was ordered to donate the *Sūtra* containers and make offerings to the gold-calligraphy copy of the *Sūtra*.[19] Murata has argued that the preparation for the Dharma assembly made by the royal family of Liang parallels the above-mentioned scene described from the narrative of Sadāprarudita, wherein the bodhisattva Dharmodgata prepared the golden tablet *Prajñāpāramitā Sūtra* and the *Sūtra* container made of gems and gave donations to them ([Murata 2020](), pp. 89–94).

Moreover, Emperor Wu did not follow the description in the narrative in all regards. He did not inscribe the *Sūtra* on a gold tablet with melted *vaiḍūrya* but on silk with Chinese sigillary characters in gold ink. This style of scripture recalls the coetaneous writing culture of the Northern Wei royal family. Murata has found that, according to the *Wei shu* 魏書, Yuan Chen 元琛 (?–526 CE) gave a gold-calligraphy copy of the *Book of Filial Piety written in Gold Calligraphy* (*Jingzi xiao jing* 金字孝經), a popular Confucian scripture, as a present to the mother of the Northern Wei emperor around 515 CE.[20] In a close period, the Royal Highness of Anfeng 安豐王, Yuan Yanming 元延明 (484–530 CE), and the Royal Highness of Zhongshan 中山王, Yuan Xi 元熙 (?–520 CE), organized a Dharma assembly for the *Huayan jing* (Ch. 華嚴經, Skt. *Avataṃsaka Sūtra*) and also created one hundred copies of that *Sūtra* together with one gold-calligraphy copy, which were all kept in the *Sūtra* containers made of four gems (cf. [Murata 2020](), pp. 77–80).[21] The veneration of the *Avataṃsaka Sūtra* in the Northern Wei was stimulated by the dissemination of the *Shidijing lun* 十地經論, the *Treatise on the Daśabhūmika Sūtra*, belonging to the *Avataṃsaka* corpus ([Tang 1983](), pp. 629–30), and therefore must have begun following the arrival of Bodhiruchi and Ratnamati in Luoyang and their translation of the *Shidijing lun* in 508 CE.

Although accounts of the gold-calligraphy scriptures from the Northern Wei predate the earliest records of the gold-calligraphy *Prajñāpāramitā Sūtra* of Empeor Wu, there is no evidence of the former directly influencing the latter. Furthermore, Emperor Wu composed the exegesis of the *Prajñāpāramitā Sūtra* in 512 CE, the eleventh year of the Tianjian regnal period 天監十一年, and, according to the *Yujiang banruojing xu* 御講般若經序 ("Preface to the Royal Lecture on the *Prajñāpāramitā Sūtra*") by Lu Yungong 陸雲公, after the composition of the exegesis, the emperor was personally engaged in preaching the *Sūtra*.[22] The gold-calligraphy copy of the *Prajñāpāramitā Sūtra* could also have been created from that date, and it would therefore be safe to say that the gold-calligraphy copy of the *Avataṃsaka Sūtra* was made around the same time, both playing a significant role in the royal Dharma assemblies and the veneration of *Sūtras* in China.

Another account concerning Emperor Wu's lecture is the "Preface to the Royal Lecture on the *Prajñāpāramitā Sūtra*" composed by Lu Yungong, who served as the secretary of the emperor, the Director of the Secretariat-Chancellery (zhongshu huangmen shilang 中書黃門侍郎), after Emperor Wu's lecture in the seventh year of the Datong regnal period 大同七年 (541 CE). Chen Jinhua has analysed the course of this Dharma assembly, pointing out that it parallels the Dharma assembly of 547 in several respects. For instance, both Dharma assemblies centred on the lecture of the *Sanhui jing* and were accompanied by a series of miraculous and propitious signs, as well as acts of self-immolation ([Chen 2006](), pp. 64–72). In particular, Lu Yungong recounts such a performance of self-immolation by a certain ascetic at the assembly:

> Shi Faxian 釋法顯 of the Aśoka Temple of Maoxian 鄞縣 in Kuaiji 會稽, who conducted ascetic practice and aspired to seek the understanding of wisdom, equalled Uttara in his thoughts and paralleled Sadāprarudita (Satuobolun 薩陀波崙, or simply Bolun 波崙) in his passionate sincerity. At the site of the lecture, after demonstrating the power of his aspiration, he cut his body and let his blood flow to the ground to express his sincerity.[23]

When listening to Emperor Wu's teaching, Fa Xian thus showed his sincerity in seeking Dharma in a manner akin to how Sadāprarudita's bodily sacrifice is described.[24] Lu Yungong further claims that the actuality of such acts of self-immolation verify the reality of the ancient stories [of Sadāprarudita and Uttara].[25] The expression, "he paralleled Sadāprarudita in his passionate sincerity" (*tong bolun zhi kendao* 同波崙之懇到), also indicates that Lu Yungong was quite acquainted with the contents of *Prajñāpāramitā Sūtra* and the religious intentions of Emperor Wu of Liang. To his mind, the monk Faxian, in coming to listen to Dharma, corresponds to bodhisattva Sadāprarudita, and Emperor Wu, in preaching the *Prajñāpāramitā Sūtra* to the Dharma preacher, bodhisattva Dharmodgata.

A similar strategy employed in Lu Yungong's preface to enhance the authority of Emperor Wu is to compare Huiling 慧令, the Great Saṅgha Rectifier 大僧正 who requested that the ruler preaches the *Prajñāpāramitā Sūtra*, with the holy disciples Subhūti and Kaśyapa, these being two prominent disciples in Mahāyāna texts who typically make requests of the Buddha to preach.[26] This indicates that Emperor Wu was likened to the Buddha himself in accepting the request of Huiling.

Additionally, Lu Yungong also drew on other elements from the narrative when describing the landscape of Hualin Park, thereby connecting the site with the scene of the narrative. Chen Jinhua has already discussed the structure and layout of the Chongyun Hall 重雲殿 in this park (Chen 2006, pp. 48–52), along with other halls within the park that had associations with Buddhism: (1) Baoyun Hall 寶雲殿 (a Buddhist Library), (2) Dengjue Hall 等覺殿, where Emperor Wu received the bodhisattva precepts in 517 CE, and (3) Huaguang Hall 華光殿, in which Emperor Wu issued his famous prohibition of the consumption of meat and alcohol within the Samgha (Chen 2005). However, we only have limited information concerning the park's landscape, except for the following passage from Lu Yungong's preface, which concerns reconstructions made by Emperor Wu:

> The Hualin Park had been the place for entertainment and banquets of the inner court since the court moved to southern China [in 317 CE] . . . After the majesty took the throne, he abandoned the entertainment. He sent back all the beautiful girls of whole palace and shared the garden with the ordinary people . . . . Through the destruction of the beautiful park, one can realize impermanence. The precious pavilion as an illusory aggregation became solid by means of the ten powers [of the Buddha]. He gave away the heavenly park and established an "Enlightenment site" (Ch. *daochang* 道場, Skt. *bodhimaṇḍa*) . . . During the springtime, all things began to grow again. The wind and the sunshine were mild and warm, neither cold nor hot. The seven rows of trees with beautiful flowers and decorated with precious substances are shining. There are ripples in the eight-fold golden pools whose beds are decorated with jade. When the gate was opened, the masses gathered. Being hurried to join the Dharma lecture, the masses were noisy. After hearing the toll of the bell, they became silent.[27]

It is worth noting that the description of the landscape of the Hualin Park in Lu Yungong's preface, such as the precious pavilion, the seven rows of trees with precious substances, and the eight pools of meritorious waters, corresponds to the city of Gandhavatī described in the narrative of Sadāprarudita. In the narrative, a Buddha image manifests in the sky and describes the city for Sadāprarudita, it being the destination of his search for the Dharma. The city is surrounded by eight pools and seven rows of trees with precious substances, and Dharmodgata preaches the *Prajñāpāramitā Sūtra* in the pavilion (Ch. *tai* 臺) or the peaked house (Skt. *kūṭāgāra*, see above note 6).

In his preface, Lu Yungong borrows the characteristic scenery of the city to sketch the landscape of the Hualin Park, albeit without mentioning Gandhavatī by name. This implies that Emperor Wu can be regarded, like the bodhisattva Dharmodgata, as the Dharma preacher of the *Prajñāpāramitā Sūtra*. Previous studies have already revealed that the landscape of Gandhavatī is similar to that of Sukhāvatī in the smaller *Sukhāvatīvyūha Sūtra*. That is to say, the Dharma preacher Dharmodgata preached the *Prajñāpāramitā Sūtra* in this city, just like Amitabha Buddha preaches in his Buddha-field of Sukhāvatī (Zhao 2020, p. 270). Thus, the description of the landscape of Hualin Park, under the influence of the narrative of Sadāprarudita, also suggests that the Dharma preacher, Emperor Wu, was regarded as having the same authority as the Buddha,[28] which is in line with the above case comparing Huiling with Subhūti and Kaśyapa.

In his record of the Chongyun Dharma assembly in 541, Lu Yungong therefore skilfully connects the scene of Emperor Wu's lecture with elements of the narrative of Sadāprarudita. His purpose, no doubt, was to affirm Emperor Wu of Liang as the Dharma preacher of the *Prajñāpāramitā Sūtra* and so at once project his religious and political authority. At the end of the preface, Lu Yungong also states, "I, an inferior minister, participated in the lecture, and my duty is to take down the historical account. I carefully recorded the current events to complete this preface",[29] taking pains to establish himself as a reliable witness of the events that transpired at the Dharma assembly. However, in utilizing the narrative of Sadāprarudita, he would also appear inclined towards exaggerating the significance of the Dharma assembly and the authority of the Dharma preacher, Emperor Wu of Liang.

To sum up the discussion thus far, we have dealt with various sources related to Emperor Wu's role in the veneration and dissemination of the *Prajñāpāramitā Sūtra*, including the ruler's own statements, in which he makes no disguise of his praise for the text and the narrative of Sadāprarudita, as well as accounts of the Dharma assemblies recorded by two witnesses, Xiao Zixian and Lu Yungong. It seems impossible that the accounts are totally independent and not influenced by Emperor Wu's personal intentions. Likewise, with regard to the record of Faxian's self-immolation, if we associate it with the fact that Emperor Wu thought highly of the Sadāprarudita narrative, it would be reasonable to assume that it was performed according to Emperor Wu's preference or even following Emperor Wu's order. However, the two accounts of the witnesses still exhibit remarkable differences in their narrative features: Xiao Zixian's account about the Dharma assembly in 533 appears to be more a statement of fact, in contrast to Lu Yungong's account about the Dharma assembly in 541, which is full of literary expression. The latter contains elements that seek to vividly recall the narrative of Sadāprarudita. The preface composed by Lu Yungong, the secretary of the emperor, could thus be regarded as a reproduction of the religious narrative within a historical document; it is a work of propaganda serving to establish the emperor's identity as a Dharma preacher of the *Prajñāpāramitā Sūtra*.

## 3. The Narrative and Master Huisi's Vow to Create the Gold-Calligraphy *Prajñāpāramitā*

The *Nanyue sidachanshi lishiyuanwen* 南嶽思大禪師立誓願文 (*Tract on the Vow Pronounced by the Great Dhyāna Master Si of Nanyue*, T. 1933; henceforth *Tract on the Vow*) is traditionally attributed to the Tiantai master Huisi 慧思 (515–577 CE). It concerns a vow to create a gold-calligraphy copy of the *Prajñāpāramitā Sūtra*. In fact, the veneration of the *Lotus Sūtra* 法華經 is also briefly mentioned in this text, but there are troubling inconsistencies concerning the two gold-calligraphy *Sūtras*. According to Stevenson and Kanno (2006, p. 89), the *Prajñāpāramitā Sūtra* is mentioned by title almost thirty times. However, the *Lotus Sūtra* appears only three times, and only two of which speak explicitly of two 二部 gold-calligraphy *Sūtras*; this claim contrasts with the large number of instances where we find explicit reference to the production of only one text, viz., the *Prajñāpāramitā Sūtra*.

Previous studies have divided the text of the *Tract on the Vow* into the following parts: (I) the Preface 序分; (II) Opening Refuges and Invocations 歸敬文; (III) Circumstances Leading to the Copying of the Sūtra 造經緣起; (IV) Main Text of Twenty-Seven Vows 誓願文; (V) Restatement of the Vows 重宣願; (VI) Petition 勸請; and (VII) Closing Injunction 後記

(Stevenson and Kanno 2006, pp. 84–86). The first-person narrative in part III reveals that the *Prajñāpāramitā Sūtra* played a significant role in the course of Huisi's dissemination of the Buddha's teaching. After his full ordination as a monk in 534 CE, Huisi studied Mahāyāna teachings and visited many meditation masters in the period of the Northern Qi 北齊. At the age of 34, he preached in Yanzhou 兗州, but his lectures were prevented when he was poisoned by some enemy monks. Subsequently, he was invited by the governor of Yingzhou 郢州 to preach the Mahāyāna teaching 摩訶衍義. After he ate a meal that had been mixed with poison by five of his rivals, on the verge of death, he confessed to the Buddhas and recollected the *Prajñāpāramitā* (*nian banruoboluomi* 念般若波羅蜜), which rid him of the poison (T. 1933: 46.787a6-b14).

This episode seems to be associated with the protective function of the *Prajñāpāramitā*, as is stated in Kumārajīva's translation of *Larger Prajñāpāramitā* (T. 223), which Master Huisi used in his lectures:

> If a son or daughter of good family hears of this profound perfection of wisdom, then holds, gets close to, recites, and correctly recollects it, and remains true to his aspiration for *sarvajñā* (omniscience), he will not be harmed by evil things, such as fumigation with poison, venom, fire or water, sword or poisoning. Why? This perfection of wisdom is the great incantation, the highest incantation.[30]

This passage in the *Larger Prajñāpāramitā* highlights that such ritual actions as reciting and recollecting the Prajñāpāramitā will not only generate immeasurable merit but also offer protection from various dangers, including poisoning, due to the *Prajñāpāramitā* being a great incantation (Skt. *mahāvidyā*, Ch. *da mingzhou* 大明呪 tr. by Kumārajīva).

In 552 CE, Master Huisi subsequently left northern China for a place of refuge in the south. When he was 42 years old (557 CE), his lectures on the *Prajñāpāramitā Sūtra* in Guangzhou 光州 were once again prevented by his rivals. Since he always came across his rivals' attack when preaching Prajñāpāramitā, he made a vow to produce a copy of the *Prajñāpāramitā Sūtra* written in gold calligraphy and kept it in a *Sūtra* container made of seven treasures 七寶經函. Then, aged 43, he was invited by the governor of Nandingzhou 南定州, a state located close to Guangzhou, to preach the teaching of Mahāyāna and was again met with certain obstacles. So, Huisi once more vowed to create a gold-calligraphy copy of the *Prajñāpāramitā Sūtra* and to make the offerings to it:

> At that time I made the vow: "I will create the gold calligraphy copy of the *Mahāyāna Prajñāpāramitā Sūtra* for all sentient beings. Pure *vaiḍūrya* and seven jewels will be used to make the *Sūtra* container that keeps the scrolls of the *Sūtra*. There will be a high couch with various precious substances, a baldachin with seven jewels, and canopies covered with pearls. All the equipment of offerings, such as flowers and necklace of precious stones, will be used as donations to the *Prajñāpāramitā Sūtra*."[31]

This passage not only mentions the gold-calligraphy copy of the *Sūtra* but also outlines the circumstances of the *Sūtra's* veneration. It shares many features with the above-mentioned passage concerning the *Sūtra's* veneration from the narrative of Sadāprarudita, and its subsequent passage explicitly refers to the main figures of the narrative, the Dharma seeker Sadāprarudita and the Dharma preacher Dharmodgata, indicating that Master Huisi's vow too was inspired by the narrative of Sadāprarudita seeking the *Prajñāpāramitā*:

> Then I shall manifest countless physical bodies for the *ṣaḍ-gati* (the "six paths" in which sentient beings are reborn), in ten directions and over innumerable *kalpas*, until awakening. I shall preach the *Prajñāpāramitā Sūtra* for all sentient beings in the ten directions. During the process, when I become a Dharma-preacher, I will be like Dharmodgata; when I become a disciple seeking the Dharma, I will be like Sadāprarudita. After making the vow, all the evil monks retreated.[32]

"Evil monks" may here refer to an arrogant rival monk, Huimiao 慧邈—following the biography of the Tiantai master Zhiyi 智顗 (538–597 CE), the *Sui Tiantai Zhizhe Dashi biezhuan* 隋天台智者大師別傳 (T. 2050), an important supplement to the *Tract on the Vow*

composed by Guanding 灌頂 (561–632 CE)—which relates the activities of the Buddhist community headed by Huisi at Mount Dasu 大蘇山 in Guangzhou. According to Guanding's account, Huisi's disciple Zhiyi defeated Huimiao in a debate over the Buddha's teaching and made him aware of the shallowness of his attainments. During the night, Zhiyi dreamed of another debate against an angry man, and he won the debate once again (Huimiao appeared in the dream in the audience). Subsequently, Zhiyi consulted with Master Huisi about his dream, and his teacher answered him by quoting the contents of Chapter 55, "Irreversibility", in the *Larger Prajñāpāramitā*, implying that Zhiyi did not waver when he faced the attacks of Māra.[33]

Guanding's account continues with the following event related to the gold-calligraphy *Larger Prajñāpāramitā*: when Huisi completed the gold-calligraphy copy, he entrusted Zhiyi to preach it. Only the concepts of "three *samādhis*" (*sansanmei* 三三昧) and the "insight of three contemplations" (*sanguanzhi* 三觀智) were open for discussion, with the remaining contents all straightforwardly declared by Zhiyi himself.[34] The *Mohe zhiguan* 摩訶止觀 (T. 1911), based on the lectures of Zhiyi and compiled by Guanding, also mentions the same fact: "Zhizhe of Tiantai preached the 'means of the doctrine' (Ch. *Famen* 法門, Skt. *dharma-paryāya*) that had been practised in his own mind … He substituted for his teacher in preaching the gold calligraphy *Prajñāpāramitā* and was respected as the teacher of emperors in the dynasties of the Chen and Sui."[35] In addition, meditation terminologies here and in Zhiyi's early monograph, the *Shi chanboluomi cidi famen* 釋禪波羅蜜次第法門 (T. 1916), are drawn extensively from the exegesis of the *Larger Prajñāpāramitā*, the *Da zhidu lun*, which indicates the significant role the *Prajñāpāramitā Sūtra* and its exegesis played in Zhiyi's early religious career.

From the dates of the above events, we can discern an underlying connection between the activities of Emperor Wu and those of Master Huisi. After the rebellion of Hou Jing 侯景, the emperor passed away in 549 CE, and his ceremonious lectures on the *Prajñāpāramitā Sūtra* came to an end. Subsequently, Master Huisi, who came to take refuge in the south and dedicated himself to preaching the *Prajñāpāramitā Sūtra*, made a vow to create a gold-calligraphy copy around 557 CE. His actions were apparently in emulation of Emperor Wu's; in addition to being acquainted with the narrative of Sadāprarudita, he even associated his own bodhisattva career in future lives with that of the bodhisattvas Sadāprarudita and Dharmodgata.

Veneration of the *Sūtra* thus extended beyond the Liang court and was a widespread phenomenon in 6th-century southern China. From the *Tract on the Vow*, it is also clear that its worship was closely integrated with certain other ideas that were prevalent in the Northern Dynasties. Unlike under Emperor Wu, as previous studies have noted, Huisi's *Sūtra* veneration was directly associated with idea of the Decline of the Dharma 末法 and the worship of the bodhisattva Maitreya, who is said to reside in Tuṣita Heaven awaiting the proper time to take his final rebirth as a Buddha. Several sections of the text express anxieties about the "Three Periods" of the True Dharma 正法, the Semblance Dharma 像法, and the Decline of the Dharma 末法, numbering 500, 1000 and 100,000 years respectively (cf. Stevenson and Kanno 2006, pp. 86–88; Murata 2020, p. 95).[36] At the end of one such section, (III) Circumstances Leading to the Copying of the Sūtra, the text specifically states that it was out of fear of decline that Huisi vowed to create the gold-calligraphy *Prajñāpāramitā Sūtra* and a container made of seven gems: "By means of the great vow, all demons and evil disasters cannot destroy it. May the future Lord Maitreya appear in the world and preach the *Prajñāpāramitā Sūtra* for countless sentient beings."[37] The *Sūtra* was thus intended for perpetuity and in anticipation of Maitreya's birth in the distant future.

These features are further confirmed by other sources related to Tiantai masters. Huisi's personal belief in Maitreya is firm and strong:

> Huisi awakened when he dreamed of the preaching of Maitreya and Amitābha, and so created statues of each and made offerings to them. He also had a dream that, following Maitreya, he joined the summit of Dragon Flower, together with the companions [of Maitreya]. He thought, "I received the *Lotus Sūtra* during

the decline of the Dharma of Śakyamuni. Now, when I came across the Lord Maitreya, I felt deeply touched and sorrowful, and suddenly awakened." He made further efforts, and auspicious miracles occurred many times.[38]

Master Huisi's belief in Maitreya and the *Prajñāpāramitā Sūtra* also influenced Zhiyi. According to Guanding's account in the *Sui Tiantai Zhizhe Dashi biezhuan* and *Xu gaoseng zhuan*, when Zhiyi was dying, he ordered his disciple Zhiyue 智越 to sweep and clean the Shicheng Temple 石城寺 so he could pass away before the stone image of Maitreya located there. Zhiyi faced west and recited the names of Amitābha and Avalokiteśvara together with the *Prajñāpāramitā*. Then, he divided his personal belongings into two portions: one to be used as a donation for Maitreya and the other to be assigned according to the *karman* 羯磨 (here denoting the rules of action within the monastic community).[39]

As Koichi Shinohara has pointed out, Zhiyi's decision to spend his final days in front of the stone image of Maitreya in Shicheng Temple was actually an incident for the Tiantai community. The stone image of Maitreya was known for its remedial capacity during Zhiyi's time, but Zhiyi failed to secure such a cure. As a result, Guanding was forced to introduce elements of the Amitābha cult to de-emphasize his failure to obtain a cure (Shinohara 1991, pp. 215–16). His recitation of the name of Prajñāpāramitā is also revealing of his attempt to be cured before his sudden death, an action that recalls the aforementioned episode from the *Tract on the Vow*, in which Huisi recollects the name of Prajñāpāramitā on the verge of death to successfully rid himself of the poison.

In comparison to Emperor Wu's lectures on the *Prajñāpāramitā Sūtra*, which served to associate him with the Dharma preacher as a means to religious authority, in Tiantai biographies, more importance was attached to the eternal aspect of the gold-calligraphy copy of the *Prajñāpāramitā Sūtra* in the context of the Decline of the Dharma, as well as to its protective function in the face of worldly dangers. Being themselves Dharma preachers of the *Prajñāpāramitā Sūtra*, Tiantai masters also drew power from the text, as Emperor Wu did, albeit under entirely different circumstances.

## 4. Conclusions

The narrative of bodhisattva Sadāprarudita seeking for *Prajñāpāramitā* was composed during the formative phases of Mahāyāna Buddhism in India. When reading through the narrative, one is first attracted by its dramatic episodes and indeed impressed by the sincerity of the Dharma seeker. The narrative's emphasis on the text's veneration and attempt at affirming the authority of the agent of the text, the Dharma preacher, would moreover appear to have been quite successful, considering that the narrative itself was always transmitted within the *Prajñāpāramitā* corpus, which is otherwise a philosophical discourse, and that it finds repeated mention throughout history.

In the 6th century CE, the narrative contributed to Chinese Buddhists' enthusiasm towards the worship and propagation of the *Prajñāpāramitā Sūtra* and its teachings. In the extant accounts of the Dharma assemblies organized by Emperor Wu of Liang, we can see that the very form of veneration described by the text, as well as the circumstances of the Dharma preacher's preaching and certain actions of the Dharma seeker, such as self-immolation, were all made to parallel scenes from the narrative. Emperor Wu of Liang himself intended to establish his religious identity as a Dharma preacher of the *Prajñāpāramitā Sūtra*, and his actions further influenced others, such as his contemporary, Master Huisi, who vowed to create the gold-calligraphy copy of *Prajñāpāramitā Sūtra*, following, it would seem, the example of Emperor Wu's Dharma assemblies.

For Chinese society of the 6th century CE, the narrative thus served as a perfect example to illustrate how the Dharma preachers and their followers may make use of religious narrative to garner veneration of Mahāyāna texts. The episodes and scenes of the Sadāprarudita narrative were constantly mentioned by the witnesses in the accounts related to Emperor Wu's Dharma assemblies and were possibly reproduced in very real ways. The underlying intention was to enhance the secular sovereign's sacredness by proclaiming his identity as a Dharma preacher of the *Prajñāpāramitā Sūtra*. However, in the biographies

of Tiantai masters, the veneration of the text, as derived from the Sadāprarudita narrative, is further integrated with the protective function of the *Prajñāpāramitā*. This rather aimed at presenting the religious leader's miraculous capability in dealing with mundane problems via the power of the *Prajñāpāramitā Sūtra*.

**Funding:** This research was funded by 國家社會科學基金重大項目：印度古典梵語文藝學重要文獻翻譯與研究 (Major project of the National Social Science Foundation of China: Indian Art and Literary Theories in Classical Sanskrit Literatures: Translation and Studies on Fundamental Works), grant number 18ZDA286.

**Data Availability Statement:** Not applicable.

**Conflicts of Interest:** The author declares no conflict of interest.

## Notes

[1] Schopen (2005) also characterizes his own early work (i.e., Schopen 1975) as a "piece of juvenilia" (Schopen 2005, p. 153, n. 118) and has provided further clarifications on this topic (Schopen 2009, 2010, 2012) (cf. Apple 2014, p. 25).

[2] He writes "since ritual [related to Mahāyāna Sūtras] involves magical power exceeding that of moral effort, the benefits are greater. The *Saddharmapuṇḍarīka-sūtra* (i.e., *Lotus Sūtra*) for example, describes the magnificent benefits that will fall on those who do no more than read, recite, copy, and uphold the *sūtra*" (Tanabe 2004, p. 532).

[3] According to the historical record, the disciple of Zhu Shixing 朱士行, Farao 法饒, brought the first Sanskrit text from Khotan. Thirty years later, it was translated by the Khotanese monk Mokṣala (Wuchaluo 無叉羅; cf. T. 2145 *Chu sanzang ji ji* 出三藏記集, *Fangguang jing ji* 放光經記).

[4] As for the brief introduction to these shorter *Sūtras*, cf. (Conze 1978, pp. 11–12).

[5] Translated from the following passage in the *Daoxing banruo jing, juan* 9: 是中有菩薩, 名曇無竭, 諸人中最高尊, 無不供養作禮者. 是菩薩用般若波羅蜜故, 作是臺. 其中有七寶之函, 以紫磨黄金為素, 書般若波羅蜜在其中, 匣中有若干百種雜名香. 曇無竭菩薩日日供養, 持雜華名香, 然燈懸幢幡. 華蓋雜寶, 若干百種音樂, 持用供養般若波羅蜜. T. 224: 8.473a21-27.

[6] For instance, the five copper plates of Helagupta made in the latter half of the first century (Salomon 2020) or the gold plate of the king of Oḍī (Swat, Pakistan), Seṇavarma, written in 14 CE (Baums 2012, p. 227).

[7] The term *kūṭāgāra* consists of *kūṭa* "peak" (or "summit") and *āgāra* "house"; literally, it therefore means "peaked house", but all the Chinese translations use the word *tai* 臺 "pavilion".

[8] Translated from the following Sanskrit passage: *tena khalu punaḥ samayena Dharmodgatena bodhisattvena mahāsattvena prajñāpāramitāyāḥ kṛtaśaḥ sapta-ratna-mayaṃ kūṭ′āgāraṃ kāritam abhūt lohita-candanālamkṛtam muktā-jāla-parikṣiptam caturṣu kūṭ′āgāra-koṇeṣu maṇi-ratnāni sthāpitāni yāni pradīpa-kṛtyaṃ kurvanti sma catasraś ca dhūpa-ghaṭikā rūpya-mayaś catur-diśam avasaktāḥ yatra śuddham kṛṣṇāguru dhūpyate sma yad uta prajñāpāramitāyāḥ pūjā'rtham tasya ca kūṭ′āgārasya madhye sapta-ratna-mayaḥ paryaṅkaḥ prajñapto 'bhūt caturṇām ratnānām pedā kṛtā yatra prajñāpāramitā prakṣiptā suvarṇa-paṭṭeṣu likhitā vilīnena vaiḍūryeṇa tac ca kūṭ′āgāram nānā-citra-paṭṭa-dāmabhiḥ pralambamānair alaṃkṛtam abhūt* (Wogihara 1932–1935, p. 955).

[9] The Chinese parallel in Kumārajīva's translation can be found in the *Xiaopin banruoboluomi jing* 小品般若波羅蜜經, *juan* 10: T. 227: 8.583b17-22.

[10] Translated from the *Gaoseng faxian zhuan* 高僧法顯傳, *juan* 1: 衆僧住處, 作舍利弗塔、目連、阿難塔, 并阿毘曇、律、經塔. 安居後一月 . . . . . . 說法已, 供養舍利弗塔. 種種香華, 通夜然燈. 使伎樂人作舍利弗大婆羅門時詣佛求出家. 大目連、大迦葉亦如是. 諸比丘尼多供養阿難塔, 以阿難請世尊聽女人出家故. 諸沙彌多供養羅云. 阿毘曇師者, 供養阿毘曇, 律師者, 供養律, 年年一供養, 各自有日. 摩訶衍人則供養《般若波羅蜜》、文殊師利、観世音等. T. 2085: 51.859b18-28.

[11] For instance, one chapter titled "The method of depicting the great Prajñāpāramitā" ("Hua dabanruoxiang fa" 畫大般若像法 in *Tuoluoni ji jing, juan* 3, T. 901: 18.805a29-c17) discusses how to depict the female bodhisattva Prajñāpāramitā and the relevant rituals in detail. Another chapter, "The method of [creating] the *maṇḍala* of Prajñāpāramitā" ("Banruo tan fa" 般若壇法 in *Tuoluoni ji jing, juan* 3, T. 901: 18.808a4-809b8) appears to further develop the description of the sacred space in T. 227, and in the Sanskrit version of the *8000 lines Prajñāpāramitā*, but the object of veneration is changed from the *Prajñāpāramitā Sūtra* to the female bodhisattva and other Buddhist gods.

[12] Nattier (2008) has also pointed out that chapters two through thirty should be considered genuine translations of Zhi Qian who revised and polished the first Chinese translation by Lokakṣema. However, the first chapter of T. 225 should not be the work either of Zhi Qian or of Kang Senghui.

[13] Modern scholarship tends to regard the *8000 lines* as the earliest version, upon which the texts of the *Larger Prajñāpāramitā* are based, whereas according to early Chinese Buddhists the former is nothing but an extract from a more comprehensive original (Zürcher 2007, p. 61)

[14] The practice of "Pure Conversation" (*qingtan* 清談) among the literati "was one of the most important factors in the spread of Buddhism in the circles of the highest gentry" (Zürcher 2007, p. 95). Particularly in the East Jin 東晉 (317–420 CE), such eminent

monks as Zhu Daoqian 竺道潛 (286–374 CE), Zhi Mindu 支愍度 and Zhi Dun 支遁 (314–366 CE), etc., were all dedicated to the preaching of the *Prajñāpāramitā Sūtra* (particularly the *Fangguang jing* 放光經).

15　Translated from the following Chinese passage: 設迺時曠正教, 處無法名, 猶且苦辛草澤, 經歷嶮遠. 翹心遏聽, 澍意希夷. 冀遲玄應, 想像空聲. 輕生以重半偈, 賣身以尊一言. 甘渫血而不疑, 欣出髓而無悋. 況復龍宮神珠, 寶臺金鍱. T. 2145: 55.54c3-7.

16　See *Xiaopin banruoboluomi jing* 小品般若波羅蜜經, *juan* 10: 其寶臺中有七寶大牀, 牀上有四寶函, 以真金鍱書般若波羅蜜, 置是函中. T. 227: 8.583b20-22.

17　The Dharma assemblies include the following: the Dharma assembly in the fifth year of the Zhongdatong regnal period 中大通五年 (533 CE) in the Temple of Tongtai 同泰寺, the assembly in the seventh year of the Datong regnal period 大同七年 (541 CE) in the Chongyun Hall, the assembly in the twelfth year of the Datong regnal period 大同十二年 (546 CE) when Emperor Wu donated his own body and the name of the era was changed to Zhong Datong 中大同 and the assembly in the second year of the Zhong Datong regnal period 中大同二年 (547 CE), when the regnal title was changed to Taiqing 太清, etc. (cf. Murata 2020, pp. 82–86).

18　Translated from the following passage of Xiao Ziliang's preface: 金字《摩訶般若波羅蜜經》者, 蓋法部之為尊 . . . . . . 皇上愛重大乘, 遨遊法藏. 道同意合, 眷懷總持. 親動王言, 妙踰綸綍 . . . . . . 乃摘以翠縑, 刻為金篆. 眾具寶飾, 品窮無價. 芝英讓巧, 金碧相輝. T. 2103: 52.236c6-13.

19　T. 2103: 52.237b22-23.

20　琛以蕭宗始學, 獻金字孝經. *Wei shu* 20:529.

21　T. 2110: 52.514c23-28.

22　上以天監十一年注釋大品, 自茲以來躬事講說. T. 2103: 52.235c28-29.

23　T. 2103: 52.236a16-19. (Cf. Chen 2006, p. 67).

24　Murata (2020, pp. 58–59) also noticed that in this passage, the name of Sadāprarudita 波崙 occurs together with that of Uttara 欝多[ 羅]. The latter refers to the story from the *Xianyu jing* 賢愚經, in which Uttara took his own bone to use as a pen and his own blood as ink when writing the teaching of the Tathāgata 如來教 (T. 202: 4.351b12-25). This story is closely connected with another object of *Sūtra* veneration that was prevalent in medieval China, the scripture written in blood ink 血字經.

25　The preface states: 昔剡体供養, 析骨书写, 归依正法, 匪悋身命. 以今望古, 信非虛说. T. 2103: 52.236a19-21.

26　大僧正慧令 . . . . . . 願等須提之問, 遂同迦葉之請. 迺啟請御講說斯經, 有詔許焉. T. 2103: 52.235c4-6.

27　See the following passage from "Preface to the Royal Lecture" collected in *Guang hongming ji* 廣弘明集, *juan* 19: 華林園者, 蓋江左以來後庭遊宴之所也 . . . . . . 自至人御宇, 屏棄聲色, 歸傾宮之美女, 共靈囿於庶人. 重以華園毀折, 悟一切之無常. 寶臺假合, 資十力而方固. 捨茲天苑, 爰建道場 . . . . . . 于時三春屆節, 萬物舒榮, 風日依遲, 不寒不暑. 瑞華寶樹, 照曜七重. 玉底金池, 淪漪八德. 洞啟高門, 雲集大眾. 趨法席以沸誼, 聽鳴鐘而寂靜. T. 2103: 52.235c8-20.

28　Skilling (2009, p. 91) has also discussed the concept of *śāstṛsaṃjñā* in *Prajñāpāramitā* texts, which equates the *Prajñāpāramitā* to the "Teacher" (i.e., the Buddha), while in the narrative of bodhisattva Sadāprarudita, the concept *śāstṛsaṃjñā* is further applied to the Dharma-preacher of the *Prajñāpāramitā Sūtra*, emphasizing that the agent of text, the Dharma-speaker, is of equal authority to the Buddha (Skilling 2009, p. 85).

29　小臣預在講筵, 職參史載. 謹錄時事, 以立今序. T. 2103: 52.236b16-17.

30　*Mohe banruoboluomi jing* 摩訶般若波羅蜜經 T. 223, *juan* 9: 是善男子、善女人聞是深般若波羅蜜, 受持、親近、讀誦、正憶念、不離薩婆若心, 若以毒藥熏、若以蠱道、若以火坑、若以深水、若欲刀殺、若與其毒, 如是眾惡皆不能傷. 何以故？是般若波羅蜜是大明呪, 是無上明呪. T. 223: 8.283b5-10.

31　Translated from the following passage from the *Tract on the Vow*: 于時發願, 我為是等及一切眾生, 誓造金字《摩訶衍般若波羅蜜》一部. 以淨瑠璃七寶作函, 奉盛經卷. 眾寶高座, 七寶帳蓋, 珠交露幔. 華香瓔珞, 種種供具 供養般若波羅蜜. T. 1933: 46.787c1-5.

32　然後我當十方六道普現無量色身, 不計劫數, 至成菩提. 當為十方一切眾生講說般若波羅蜜經. 於是中間, 若作法師如曇無竭, 若作求法弟子如薩陀波崙. 發願之後, 眾惡比丘皆悉退散. T. 1933: 46.787c5-9.

33　Cf. T. 2050: 50.192a8-22.

34　Cf. T. 2050: 50.192a23-25.

35　According to Guanding's statement in *Mohe zhiguan*, *juan* 1: 天台智者說己心中所行法門 . . . . . . 代受法師講金字般若, 陳隋二國宗為帝師. T. 1911: 46.1b13-16.

36　As Chen Zhiyuan has shown, in medieval China, two methods of counting the Buddha's birth year were accepted: the tenth year under the rule of King Zhuang of Zhou Dynasty 周莊王十年, which was adopted by historians from the Southern Dynasties to the Sui 隋, and the twenty-fourth year under the rule of the King Zhao of Zhou Dynasty 周昭王二十四年, which was prevalent in the Northern Qi 北齊. Huisi's opinion falls under the latter (Chen 2018, p. 126).

37　Translated from the following passage from *Tract on the Vow*: 以大願故, 一切眾魔諸惡災難不能沮壞. 願於當來彌勒世尊出興于世, 普為一切無量眾生說是般若波羅蜜經時. T. 1933: 46.787c22-25.

38　According to the *Xu gaoseng zhuan* 續高僧傳, *juan* 17: 又夢彌勒、彌陀說法開悟, 故造二像並同供養, 又夢隨從彌勒與諸眷屬同會龍華. 心自惟曰：" 我於釋迦末法受持法華, 今值慈尊, 感傷悲泣, 豁然覺悟, " 轉復精進, 靈瑞重沓, T. 2060: 50.562c21-25.

39　Cf. T. 2060: 50.567b25-29; T. 2050: 50.196a9-14.

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
