# Peer review of "The Story of Sadāprarudita’s Search for Dharma and the Worship of the Prajñāpāramitā Sūtra from India to Sixth-Century China"

_religions, doi:10.3390/rel14030410_

Round 1
Reviewer 1 Report
This is a fascinating, although VERY poorly prepared, academic article that demonstrates some excellent research by a scholar reading primarily Chinese editions of Prajñāpāramitā literature. With some slight attention to context and especially about why the research question is so important to the study of Mahāyāna Buddhist literature not already conducted by Schopen (as the author states), this could become a very good academic article. But heavy editing is required to provide context for readers who may not be very familiar with both Chinese translations of Buddhist texts and how to find and cite them.
The entire paper requires thorough editing to correct mistakes like Ch. (for Chinese) on p. 2. On p.3 the paragraph that begins with "The prototype..." needs to be indented. Figures including Faxian (p. 4) require dates when known. ALL Prajñāpāramitā texts cited (especially less well-known ones to readers unfamiliar with Conze's pioneering work in this area) need to be introduced and briefly explained. On p. 6, the Fangguang jing 放光經 is an example of many, many.
Footnotes cannot be only in Chinese, even if the original text translated in the body of the paper appears here. Titles WITH PINYIN Romanization and juan 卷 number MUST be added throughout. See footnotes 14 (p. 7), 17-21 (p. 9), 22, 24, 25 (p. 10), 26, 28 (p. 11), 29 (p. 12), and so on and so forth. Otherwise the purpose of these footnotes remains unknown to most readers.
Author Response
Point 1: The entire paper requires thorough editing to correct mistakes like Ch. (for Chinese) on p. 2. On p.3 the paragraph that begins with "The prototype..." needs to be indented. Figures including Faxian (p. 4) require dates when known. ALL Prajñāpāramitā texts cited (especially less well-known ones to readers unfamiliar with Conze's pioneering work in this area) need to be introduced and briefly explained. On p. 6, the Fangguang jing 放光經 is an example of many, many.
Response 1: I made thorough editing to revise the mistakes mentioned in Point 1.
Footnotes cannot be only in Chinese, even if the original text translated in the body of the paper appears here. Titles WITH PINYIN Romanization and juan 卷 number MUST be added throughout. See footnotes 14 (p. 7), 17-21 (p. 9), 22, 24, 25 (p. 10), 26, 28 (p. 11), 29 (p. 12), and so on and so forth. Otherwise the purpose of these footnotes remains unknown to most readers.
Response 2: I added some information to the footnotes to make it clearer.
Reviewer 2 Report
“The Story of Sadāprarudita’s Search for the Prajñāpāramitā and the Worship of the Prajñāpāramitā Sūtra across Cultures” is a cogently argued essay for the most part. It advances an argument regarding the influence of the story of Sadāprarudita and the family of Prajñāpāramitā sutras in early medieval China, particularly due to the patronage of Liang emperor Wu (r. 502–549). However, stylistically, the essay needs much editorial revision. In addition, the paper should probably be retitled because “across Cultures” is vague. The cultures referred to in this paper are, generally stated, “medieval India and China”—although the information on India pales in comparison to that of sixth-century China. The author should consider changing the title to “The Story of Sadāprarudita’s Search for the Prajñāpāramitā and the Worship of the Prajñāpāramitā Sūtra in Sixth-Century China” or perhaps “The Story of Sadāprarudita’s Search for the Prajñāpāramitā and the Worship of the Prajñāpāramitā Sūtra in Early Medieval India and China.”
The stylistic problems in this essay are manifold: They range from romanization issues (not following the rules of pinyin) to lack of consistency in how titles are treated (sometimes an English title is used, sometimes a Sanskrit title is used for a text in Chinese, and sometimes a Pinyin title is used). Several block quotes are not marked and indented as block quotes. In addition, the author does not differentiate between different levels of material: titles of prefaces to sūtras and the names of chapters within sūtras should be placed in quotation marks; and stand-alone texts, like sūtras and vow texts (yuanwen) should be italicized. On the whole, this essay should be published after a thorough vetting of the style so that it is consistent with the rest of the paper and with the journal style.
This reviewer has included a marked/corrected pdf version that the author in encouraged to consult.
The author should be consistent in the way the author refers to different versions of the Perfection of Wisdom.
ln. 49: “Perfect of Wisdom in 8000 Lines” should be Perfection of Wisdom in 8,000 Lines
ln. 65: Daoxing pin should be “Daoxing pin” because it is a chapter of the Bore jing.
ln. 65: The sinographs 般若 are pronounced bore in modern Mandarin, so why not romanize them as bore instead of banruo? Daoxin banruo jing should be Daoxin bore jing.
ln. 75, 79-80, 89: 8000 lines; does this refer to the Perfection of Wisdom in 8,000 Lines as a text? If so, then italicize (8,000 Lines) or, even better, use the full name.
ln. 118-119: 25000 lines Prajñāpāramitā should be Perfection of Wisdom in 25,000 Lines
ln. 170-180: 25,000 lines Prajñāpāramitā should be Perfection of Wisdom in 25,000 Lines
p. 3 n. 6: Regarding the sinograph tai 臺, although “deck” or “terrace” is a primary meaning of tai, the term does appear in the names of buildings in medieval Chinese texts as a multistoried pavilion or tower (louge 樓閣) or hermitage or monastery (jingshe 精舍). See, for instance, how the term is used in the Gaoseng zhuan and Xu gaoseng zhuan.
ln. 264-265: the theory of Buddhadhātu 佛性 should be the theory of Buddha nature 佛性. Technically speaking, 佛性 is Buddhatā; but in Anglophone Buddhist studies, it is always referred to as Buddha nature.
ln. 273: Preface to the Exegesis for the Larger Prajñāpāramitā Sūtra 註解大品序 should be “Preface to the Exegesis on the Larger Prajñāpāramitā Sūtra” (“Zhujie Daping xu” 註解大品序)
ln. 350-351: Colophon of the Royal Lecture of Mahāprajñāpāramitā Sūtra written in Gold Calligraphy 御講金字摩訶般若波羅蜜經序 should be “Preface to the Mahāprajñāpāramitā Sūtra written in Gold Letters for use in Royal Lectures” (“Yujiang jinzi Mohe bore boluomijing xu” 御講金字摩訶般若波羅蜜經序)
ln. 378-379: Jingzi xiao jing 金字孝經 should be Book of Filial Piety written in Gold Letters (Jingzi xiao jing 金字孝經). The Book of Filial Piety is the popular Confucian classic. The written in gold letters shows that it was likely for imperial use.
ln. 394-395: Preface to the Royal Lecture of the Prajñāpāramitā Sūtra by Lu Yungong has some problems. It should be “Preface to the Royal Lecture on the Prajñāpāramitā Sūtra” (“Yujiang Bore jing xu” 御講波若經序) by Lu Yun
ln. 425, 440, 463: Preface to the Royal Lecture should be “Preface to the Royal Lecture”
ln. 395, : Lu Yungong 陸雲公. This reviewer can only find the name Lu Yun 陸雲 in the Guang hongming ji. However, Lu Yungong appears in the Liang shu and other texts? Please check.
ln. 402-403: the attendant or the secretary of emperor in the central drafting office 中書黃門侍郎 should be Director of the Chancellery-Secretariat (zhongshu huangmen shilang 中書黃門侍郎)
p. 9, n. 19: This note should cite the Zhonghua shuju edition of the Wei shu: Wei shu 20:529.
For bibliography: Wei shu 魏書 (History of the Wei). 114 rolls. Compiled by Wei Shou 魏收 (506–572) between 551–554. 8 vols. Beijing: Zhonghua shuju, 1974.
ln. 448: bodhimaṇḍa 道場 (“the terrace of enlightenment”) should be bodhimaṇḍa 道場 (enlightenment site). In medieval China, however, daochang was also used as a common term for a monastic complex. “Enlightenment site” (daochang 道場) was a common term for monastery during the seventh century. It was originally a translation of bodhimaṇḍa, the place under the bodhi tree where Śākyamuni became enlightened. During the Northern Wei period (386–535) it came to mean a place where Buddhist rituals and ceremonies, such as ordinations and special assemblies, were performed. Sui emperor Yang (r. 604–617) then changed the name for all monasteries in China to “enlightenment site” in 613. It is also likely that during the period of Empress Wu’s influence and reign (ca. 660–705) the term was again substituted for the ordinary term for monastery (si) given that the imperial temples in the eastern and western palaces bore the designation interior enlightenment site (neidaochang 內道場).
ln. 543 and p. 12, n. 29: a mahāvidyā is a damingzhou 大明呪; because ming 明are vidyā. Da zhoushu 大呪術 is a related term, but zhoushu 呪術 is a term usually associated with dhāraṇī techniques and sometimes best translated as “spell-craft.”
Bibliography
The author should provide bibliographic information for numerous texts contained in the Taishō, which are cited in the paper:
T. 207
T. 223
T. 224
T. 225
T. 227
T. 901
T. 1509
T. 1911
T. 1933
T. 2050
T. 2060
T. 2085
T. 2103
T. 2110
T. 2121
T. 2145

Author Response
Point 1: The paper should probably be retitled because “across Cultures” is vague. The cultures referred to in this paper are, generally stated, “medieval India and China”—although the information on India pales in comparison to that of sixth-century China. The author should consider changing the title to “The Story of Sadāprarudita’s Search for the Prajñāpāramitā and the Worship of the Prajñāpāramitā Sūtra in Sixth-Century China” or perhaps “The Story of Sadāprarudita’s Search for the Prajñāpāramitā and the Worship of the Prajñāpāramitā Sūtra in Early Medieval India and China.”
Response 1: I changed the title to: The Story of Sadāprarudita’s Search for the Dharma and the Worship of the Prajñāpāramitā Sūtra from India to Sixth-Century China.
Point 2: The stylistic problems in this essay are manifold: They range from romanization issues (not following the rules of pinyin) to lack of consistency in how titles are treated (sometimes an English title is used, sometimes a Sanskrit title is used for a text in Chinese, and sometimes a Pinyin title is used). Several block quotes are not marked and indented as block quotes. In addition, the author does not differentiate between different levels of material: titles of prefaces to sūtras and the names of chapters within sūtras should be placed in quotation marks; and stand-alone texts, like sūtras and vow texts (yuanwen) should be italicized. On the whole, this essay should be published after a thorough vetting of the style so that it is consistent with the rest of the paper and with the journal style...
Response 2: I corrected the mistakes and inconsistencies thanks to the detailed suggestion in the review.